# Quantum Monte Carlo simulation of the 3D Ising transition on the fuzzy sphere

Johannes Stephan Hofmann[1⋆], Florian Goth[2], Wei Zhu[3],
Yin-Chen He[4] and Emilie Huffman[4†]

**1** Department of Condensed Matter Physics, Weizmann Institute of Science, Rehovot, 76100, Israel
**2** Institute for Theoretical Physics, University of Würzburg, Würzburg, 97074, Germany
**3** School of Science, Westlake University, Hongzhou, 310024, P. R. China
**4** Perimeter Institute for Theoretical Physics, Waterloo, Ontario N2L 2Y5, Canada

⋆ johannes-stephan.hofmann@weizmann.ac.il , † ehuffman@perimeterinstitute.ca

## Abstract

We present a numerical quantum Monte Carlo (QMC) method for simulating the 3D phase transition on the recently proposed fuzzy sphere [1]. By introducing an additional $SU(2)$ layer degree of freedom, we reformulate the model into a form suitable for sign-problem-free QMC simulation. From the finite-size-scaling, we show that this QMC-friendly model undergoes a quantum phase transition belonging to the 3D Ising universality class, and at the critical point we compute the scaling dimensions from the state-operator correspondence, which largely agrees with the prediction from the conformal field theory. These results pave the way to construct sign-problem-free models for QMC simulations on the fuzzy sphere, which could advance the future study on more sophisticated criticalities.

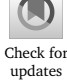

# 1  Introduction

Critical phenomena emerging at classical and quantum phase transitions are of great interest due to their experimental relevance and theoretical significance [2, 3]. Many critical phenomena are believed to be described by conformal field theories (CFTs), which are strongly-interacting and pose challenges for studies in higher space-time dimensions beyond 2D (i.e., 1+1D). A recent method known as fuzzy (non-commutative) sphere regularization [1] has been invented to investigate 3D (i.e., 2+1D) critical phenomena governed by 3D CFTs on a cylindrical geometry represented as $S^2 \times \mathbb{R}$. Compared to traditional lattice regularization, the fuzzy sphere regularization offers numerous advantages in the study of 3D CFTs, primarily due to the utilization of radial quantization in $S^2 \times \mathbb{R}$ [4, 5] as well as the exact preservation of sphere $SO(3)$ symmetry [6, 7], as convincingly demonstrated recently [1, 8–11].

Firstly, the fuzzy sphere enables direct access to information regarding the emergent conformal symmetry in the critical state [1, 10]. Secondly, it allows for the direct extraction of various data of the CFTs, including numerous scaling dimensions of conformal primary operators [1, 10], operator product expansion coefficients [8], and four-point correlators [9]. For instance, scaling dimensions can be computed directly from excitation energies of the system, and their accuracy can be further improved using conformal perturbation [12]. Thirdly, the fuzzy sphere scheme is applicable to a variety of 3D CFTs, including Ising [1], O(N) Wilson-Fisher, SO(5) deconfined phase transition [10], critical gauge theories [10], and defect CFTs [11]. Lastly, the fuzzy sphere regularization exhibits an incredibly small finite-size effect when the Hamiltonian is reasonably fine-tuned. These advantageous features of fuzzy sphere regularization present an exciting opportunity to explore 3D CFTs with high efficiency, accuracy, and comprehensiveness.

The fuzzy sphere regularization considers a microscopic quantum Hamiltonian modeling fermions (with multiple flavors) on continuous spherical space and projecting fermions into the lowest spherical Landau level [1, 6, 13]. In comparison with the regular lattice model, the fuzzy sphere model preserves the continuous rotational symmetry exactly in the UV limit. Thanks to the extremely small finite-size effect achieved through fine-tuning, numerical algorithms such as exact diagonalization (ED) and density matrix renormalization group (DMRG) methods are highly effective in studying the fuzzy sphere model of the 3D Ising CFT and SO(5) deconfined phase transition. However, the computational cost of these two algorithms will eventually grow exponentially with the system size. More importantly, for cases involving a large number of fermion flavors, the computational costs of ED and DMRG quickly surpass practical resource and time limitations. In these cases, it would be helpful to be able to study models on the fuzzy sphere using a method that scales polynomially in time, such as quantum Monte Carlo (QMC).

The aim of this paper is to utilize the 3D Ising CFT as an example to demonstrate the application of the QMC approach in studying 3D CFTs on the fuzzy sphere. A similar discussion for the fuzzy torus model can be found in Ref. [13, 14]. In contrast to the fuzzy sphere Ising model introduced in Ref. [1], we introduce an additional flavor index to the fermions, which results in no sign problem for the QMC simulations. As a benchmark, we provide numerical

results of finite-size scaling, indicating that this model also belongs to the 3D Ising universality class. Furthermore, we introduce observables that enable the extraction of energy gaps in the spectrum corresponding to specific symmetry quantum numbers. This allows us to investigate the presence of conformal symmetry at criticality and extract scaling dimensions through the state-operator correspondence. Our numerical results for energy gaps are consistent with the universality of the 3D CFT Ising model, albeit with a larger finite-size effect compared to the previously studied fuzzy sphere Ising model [1]. In summary, we believe the QMC enriches the arsenal to study the fuzzy sphere model.

This paper is organized as follows: in Section II we introduce the model and its symmetries, and we discuss how it can be implemented in auxiliary-field QMC simulations. We also argue for why the simulations are sign-problem-free. In Section III we discuss finite-size-scaling results and give evidence that the model is in the 3D Ising universality class, and we discuss energy spectrum results and give evidence for emergent conformal symmetry. Section IV contains our conclusions.

## 2 Model and method

### 2.1 Review of fuzzy sphere regularization

The fuzzy sphere regularization considers fermions moving on a sphere in the presence of a magnetic monopole with $4\pi s$ flux sitting in the center of the sphere. In general, we can consider multi-flavor fermions $\psi_\alpha$ with the flavor index $\alpha$, described by a Hamiltonian,

$$H = H_{\text{kin}} + H_{\text{int}}. \tag{1}$$

Here $H_{\text{kin}}$ is the kinetic term of fermions, and $H_{\text{int}}$ is an interaction which takes forms such as a density-density interaction,

$$\int d^2\vec{r}_1 d^2\vec{r}_2 \, U(\vec{r}_1 - \vec{r}_2) n^a(\vec{r}_1) n^b(\vec{r}_2), \tag{2}$$

where $n^a(\vec{r}) = \psi^\dagger(\vec{r})_\alpha \psi(\vec{r})_\beta M^a_{\alpha\beta}$ and $M^a$ is a matrix defined in the fermion flavor space. $U(\vec{r}_1 - \vec{r}_2)$ is a rotationally invariant interaction, and we take it to be short ranged such as $\delta(\vec{r}_1 - \vec{r}_2)$ and $\nabla^2\delta(\vec{r}_1 - \vec{r}_2)$.

The energy levels of $H_{\text{kin}}$ form quantized Landau levels, whose wave-functions are described by the monopole Harmonics (i.e. spin-weighted spherical Harmonics) $Y^{(s)}_{n+s,m}(\theta, \varphi)$ [15], with $n = 0, 1, \cdots$ as the Landau level index and $(\theta, \varphi)$ as the spherical coordinates. Each Landau level has an energy $E_n = [(n + 1/2) + n(n + 1)/2s]\omega_c/2\pi$, with the cyclotron frequency $\omega_c$ [6, 16]. The states of each Landau level have an angular momentum $L = s + n$, hence they are $(2s + 2n + 1)$-fold degenerate, which can be labeled by the quantum number of the $z$-component of the angular momentum $L^z$, $m = -s - n, -s - n + 1, \cdots, s + n$. Because we may set the scale of $H_{\text{kin}}$ as large as we like ($\omega_c \to \infty$) relative to our scale of interest for $H_{\text{int}}$, we may enforce that there are no fluctuations out of the lowest Landau level (LLL). The physics of interest will come from the terms that make up $H_{\text{int}}$. Hence, we consider the limit that $H_{\text{kin}} \gg H_{\text{int}}$ such that we can project the system into the LLL. The annihilation operator $\psi(\theta, \varphi)$ on the LLL can be written as

$$\psi_\alpha(\theta, \varphi) = \frac{1}{\sqrt{N}} \sum_{m=-s}^{s} \bar{Y}^{(s)}_{s,m}(\theta, \varphi) c_{m,\alpha}, \tag{3}$$

where $c_{m,\alpha}$ stands for the annihilation operator of Landau orbital $m$, and it is independent of coordinates $(\theta, \varphi)$. $N = 2s + 1$ is the number of orbitals, playing the role of area of the 2D

space. The prefactor $1/\sqrt{N}$ ensures that the density operator,

$$n^a(\theta, \varphi) = \frac{1}{N} \sum_{m_1, m_2} Y_{s,m_1}^{(s)} \bar{Y}_{s,m_2}^{(s)} c_{m_1,\alpha}^\dagger c_{m_2,\beta} M_{\alpha\beta}^a, \tag{4}$$

is an intensive quantity.

Under this LLL projection Eq. (3), any rotation invariant density-density interaction in the form of eq. (2) can be written as the Haldane pseudopotentials [6] in terms of second quantized fermion operators,

$$\sum_{m_1, m_2, m_3, m_4} V_{m_1, m_2, m_3, m_4} (c_{m_1,\alpha}^\dagger M_{\alpha\beta}^a c_{m_4,\beta})(c_{m_2,\eta}^\dagger M_{\eta\gamma}^b c_{m_3,\gamma}), \tag{5}$$

with

$$V_{m_1, m_2, m_3, m_4} = \delta_{m_1 + m_2, m_3 + m_4} \sum_{l=0}^{2s} V_l \, (4s - 2l + 1)$$

$$\times \begin{pmatrix} s & s & 2s-l \\ m_1 & m_2 & -m_1 - m_2 \end{pmatrix} \begin{pmatrix} s & s & 2s-l \\ m_4 & m_3 & -m_3 - m_4 \end{pmatrix}, \tag{6}$$

where $\begin{pmatrix} j_1 & j_2 & j_3 \\ m_1 & m_2 & m_3 \end{pmatrix}$ is the Wigner 3$j$-symbol. $V_l$ are numbers whose values are specifically depending on the form of the interaction $U(\vec{r}_1 - \vec{r}_1)$. For the remainder of this work, we focus on $U(\vec{r}_1 - \vec{r}_1) = U(\Omega_{12}) = \frac{g_0}{N} \delta(\Omega_{12}) + \frac{g_1}{N^2} \nabla^2 \delta(\Omega_{12})$ with $\delta(\Omega_{12}) = \delta(\varphi_1 - \varphi_2)\delta(\cos\theta_1 - \cos\theta_2)$. The corresponding Haldane pseudo-potentials are

$$V_0 = \frac{2s+1}{4s+1} g_0 - \frac{s}{4s+1} g_1, \qquad V_1 = \frac{s}{4s-1} g_1, \qquad V_{l \geq 2} = 0. \tag{7}$$

To realize the $2 + 1$D Ising transition, Ref. [1] introduced a Hamiltonian that has two flavors of fermions $\psi^\dagger = (\psi_\uparrow^\dagger, \psi_\downarrow^\dagger)$ with their interaction,

$$H_{\text{int}} = \int N^2 \, d\Omega_1 d\Omega_2 \, U(\Omega_{12}) \big[ n^0(\theta_1, \varphi_1) n^0(\theta_2, \varphi_2) - n^z(\theta_1, \varphi_1) n^z(\theta_2, \varphi_2) \big]$$

$$- h \int N \, d\Omega \, n^x(\theta, \varphi), \tag{8}$$

where $\Omega = (\theta, \varphi)$ is a spherical coordinate and $n^a(\theta, \varphi) = \psi^\dagger(\theta, \varphi)\sigma^a \psi(\theta, \varphi)$ is a local density operator with $\sigma^{x,y,z}$ being Pauli matrices, $\sigma^0 = I_{2\times2}$. The first term behaves like an Ising ferromagnetic interaction, while the second term is the transverse field.

It is straightforward to solve the second quantized Hamiltonian Eq. (5) using unbiased numerical algorithm such as the ED and the DMRG, although their computational costs grow exponentially with the system size $N = 2s + 1$. So it is highly desirable to develop QMC method for the simulation of a fuzzy sphere model, and it is the focus of this paper.

It is worth mentioning why the LLL projection leads to a fuzzy sphere. We can consider the projection of the coordinates of a unit sphere, denoted as $\boldsymbol{x} = (\sin\theta\cos\varphi, \sin\theta\sin\varphi, \cos\theta)$. After the projection, the coordinates are transformed into $(2s + 1) \times (2s + 1)$ matrices, where $(\boldsymbol{X})_{m_1, m_2} = \int d\Omega \, \boldsymbol{x} \, \bar{Y}_{s,m_1}^{(s)}(\Omega) Y_{s,m_2}^{(s)}(\Omega)$. These matrices satisfy the following relations:

$$[X_i, X_j] = \frac{1}{s+1} i\epsilon_{ijk} X_k, \qquad \sum_{i=1}^{3} X_i X_i = \frac{s}{s+1} \mathbf{1}_{2s+1}. \tag{9}$$

The fact that the three coordinates satisfy the $SO(3)$ algebra formally defines a fuzzy sphere [7]. It is interesting to note that in the limit as $s \to \infty$, the fuzziness disappears and a unit sphere is recovered.

## 2.2 The density form of interaction

To facilitate QMC simulation, we would like to write the Hamiltonian in terms of the density operator in the angular momentum space $n_{l,m}^a$, defined as,

$$n^a(\theta,\varphi) = \frac{1}{N} \sum_{l,m} n_{l,m}^a Y_l^m(\theta,\varphi).\tag{10}$$

Here $Y_l^m(\theta,\varphi)$ is the spherical harmonics, with $m = -l, -l+1, \cdots, l$ and $l \in \mathbb{Z}$. $n_{l,m}^a$ can be obtained using the spherical harmonic transformation,

$$
\begin{aligned}
n_{l,m}^a &= N \int d\Omega \, \bar{Y}_l^m(\theta,\varphi) n^a(\theta,\varphi) \\
&= N \sqrt{\frac{2l+1}{4\pi}} \sum_{m_1=-s}^{s} (-1)^{3s+m_1} \begin{pmatrix} s & l & s \\ -m_1 & m & m_1-m \end{pmatrix} tjsls-s0s c_{m_1,\alpha}^\dagger c_{m_1-m,\beta} M_{\alpha\beta}^a.
\end{aligned}\tag{11}
$$

To have the term $\begin{pmatrix} s & l & s \\ -m_1 & m & m_1-m \end{pmatrix}$ non-vanishing, we should have $l \le 2s$. One can show $n_{l,m}^\dagger = (-1)^m n_{l,-m}$.

In this context, it is convenient to decompose the potential $U(\theta_{12}) = \sum_l \frac{2l+1}{4\pi} U_l P_l(\cos\theta_{12})$ using the Legendre polynomials, $P_l(\cos\theta_{12}) = \frac{4\pi}{2l+1} \sum_{m=-l}^{l} \bar{Y}_l^m(\Omega_1) Y_l^m(\Omega_2)$, such that the interaction terms take the form

$$\int N^2 d\Omega_1 d\Omega_2 U(\theta_{12}) n^a(\theta_1,\varphi_1) n^b(\theta_2,\varphi_2) = \sum_{l=0}^{2s} U_l \sum_{m=-l}^{l} (n_{l,m}^a)^\dagger n_{l,m}^b,\tag{12}$$

with the coefficients $U_l = g_0/N - l(l+1)g_1/N^2$.

## 2.3 Four component fuzzy sphere model

In comparison to Ref. [1], we consider four flavors of fermions, $\psi^\dagger = (\psi_{\uparrow,+}^\dagger, \psi_{\uparrow,-}^\dagger, \psi_{\downarrow,+}^\dagger, \psi_{\downarrow,-}^\dagger)$, i.e., we introduce an additional "layer" degree of freedom $(+,-)$. The Pauli-matrices $\sigma^i$ and $\tau^i$ act on the spin $(\uparrow,\downarrow)$ and layer indices, respectively. Let us define the operators $n_{l,m}^0$ and $n_{l,m}^z$ according to Eq. (11) with $M^0 = \sigma^0\tau^0$ and $M^z = \sigma^z\tau^0$, respectively. The Hamiltonian reads

$$H_{\text{int}} = \sum_{l=0}^{2s} U_l \sum_{m=-l}^{l} \left[ (n_{l,m}^0)^\dagger n_{l,m}^0 - (n_{l,m}^z)^\dagger n_{l,m}^z \right] + h \sum_m c_m^\dagger \sigma^x \tau^0 c_m,\tag{13}$$

and the interaction favors a ferromagnetic state for $g_0, g_1 > 0$.

There are four symmetries of this model, which are

1. Ising $\mathbb{Z}_2$ symmetry: $c_m \to \sigma^x\tau^0 c_m$.

2. $SO(3)$ sphere rotation symmetry: $c_{m=-s,\dots,s}$ form the spin-$s$ representation of $SO(3)$.

3. Particle-hole symmetry: $c_m \to i\sigma^y\tau^0 c_m^\dagger$, and $i \to -i$.

4. Layer SU(2): generated by $c_m \to \sigma^0\tau^{x,y,z} c_m$.

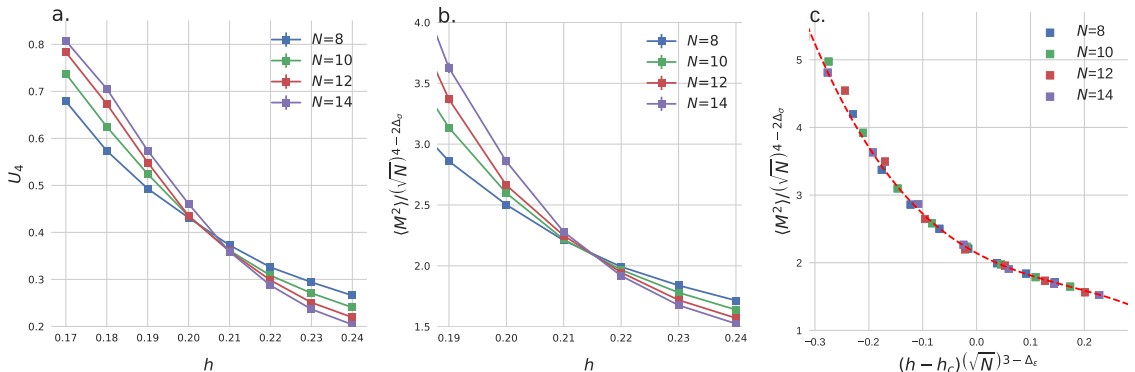

Figure 1: Order parameter data for $V_0 = 0.5564$ and $V_1 = 0.1$, which shows evidence for a continuous phase transition consistent with that of the 3D Ising Model. (a) Binder ratio data showing a crossing that drifts somewhat in system size: $N = 12$ and $N = 14$ cross around $h = 0.21$, whereas $N = 8$ and $N = 10$ cross closer to $h = 0.20$. (b) Magnetization data showing consistent crossing between $h = 0.21$ and $h = 0.22$ for $N = 10, 12, 14$. Ising $\Delta_\sigma = 0.518$ is assumed. (c) Magnetization data plotted along with a universal scaling function fit. Fixing $\Delta_\sigma = 0.518$ and $\Delta_\epsilon = 1.41$ (as shown in this figure) yields a good fit ($\chi^2 = 1.305$) with $h_c = 0.2129(8)$, consistent with 3D Ising universality. Fitting using an $h = 0.21$ estimate consistent with the Binder ratio and magnetization crossing data gives $\Delta_\sigma = 0.49(2)$ and $\Delta_\epsilon = 1.28(6)$.

The first three of these symmetries are the same as those of the two-flavor model studied in [1]. The layer symmetry is an additional symmetry for the four flavors, which allows for sign-problem-free QMC simulations of this model. At the Ising transition, the layer $SU(2)$ degrees of freedom need to be gapped. We have verified this in Appendix C.

Before moving on, we remark that the four component fuzzy sphere model Eq. 13 is not a simple product of the two-component model Eq. 8, so the phase transition point $h_c$ of the four component model is distinct from that of the two-component model. While the interaction is chosen to be of density form in the layer degree of freedom, i.e., using $\tau^0$, to disfavor layer-magnetism, the two layers are coupled and spontaneous layer-symmetry breaking cannot ruled out in general. Nevertheless, we ensure the universality of the four component model falls in the 3D Ising class, as shown below.

## 3  Results

### 3.1  QMC simulations

We simulate the model, eq. (13), using projector auxiliary Quantum Monte Carlo (AFQMC). To fit this goal, we rewrite the sums of quartic terms in the following way

$$U_l \sum_{m=-l}^{l} \left(n_{l,m}^a\right)^\dagger n_{l,m}^a = U_l \sum_{m=-l}^{l} (-1)^m \left(n_{l,-m}^a\right) n_{l,m}^a$$

$$= \frac{U_l}{4} \sum_{m=-l}^{l} \left((1+i)n_{l,m}^a + (1-i)(-1)^m n_{l,-m}^a\right)^2, \tag{14}$$

where in the first equality we make use of the density operator identity

$$n_{l,m}{}^\dagger = (-1)^m n_{l,-m}. \tag{15}$$

The squared operators in the second line of (14) are Hermitian, and thus AFQMC as implemented in [17] is applicable. The projector we use is the half-filled solution to the model when $g_0 = g_1 = 0$, where the Ising spins are polarized by the transverse field term $h \sum c_m^\dagger \sigma^x \tau^0 c_m$.

Now we show, that the QMC simulation of this model is sign-problem-free. After the Hubbard-Stratonovich transformation, we have a prefactor of $\sqrt{-\Delta \tau U_l / 4}$. If $g_0 - g_1 l(l+1)/(2s+1)$ of the expression in (13) is always positive, then we get an extra factor of $i$ for the $n^0$ terms, which picks up a sign under antiunitary transformations. The antiunitary particle-hole transformation $\mathcal{P}$,

$$
\begin{aligned}
c_m^\dagger &\to i\sigma^y \tau^0 c_m, \quad i \to -i, \\
n_{l,m}^z &\to (-1)^m n_{l,-m}^z, \\
n_{l,m}^0 &\to -(-1)^m n_{l,-m}^0,
\end{aligned}
\tag{16}
$$

is a symmetry with $\mathcal{P}^2$. Combined with the $SU(2)$ layer symmetry, it guarantees the absence of the sign-problem in this model [18]. Hence, the computational complexity scales polynomially with system size $N$. However, in this basis, the auxiliary fields couple to the operators $n_{l,m}^a$ of typical rank $N$, compared to the usual rank of $\mathcal{O}(1)$ in lattice models. Therefore, the compute time of the algorithm scales as $N^4$ instead of the conventional $N^3$.

In this particular model, we focus on the critical point that occurs in a regime where both $g_0, g_1 > 0$. We have not proven in the discussion above that the absence of a sign problem occurs when instances of $g_0 - g_1 l(l+1)/(2s+1)$ is positive for small $l$ but is negative for large $l$, yet we encountered no sign problem in our simulations. One explanation may be that the Wigner-3j prefactor $\begin{pmatrix} s & l & s \\ -s & 0 & s \end{pmatrix}$ decays exponentially in $l$, causing a suppression of terms that change the overall prefactor signs in (13), and so the smallness of the couplings of these terms may be important.

Due to the nonlocal nature of the operators in (13), controlling Trotter discretization errors becomes a more demanding task, as observed in [14]. We alleviate some of these effects by adopting a stabilized second-order Trotter decomposition developed by Blanes et al., as discussed in [19]. The effectiveness of these alternate splitting schemes in the realm of AFQMC was shown in [20]. Furthermore, to implement the Wigner-3j prefactors efficiently, we utilize the software package detailed in [21].

Here we utilize QMC to compute the evolution of the order parameter and CFT dimensionless two-point correlators across the transition point, as well as extract energy gaps for the lowest lying states using time-displaced correlation functions (see Appendix B). While the Lowest Landau level basis has already been used for a QMC study in [22], the CFT-inspired use of time-displaced correlation functions and two-point correlators on the fuzzy sphere is new for QMC studies.

Below we will set $V_1 = 0.1$, $V_0 = 0.5564$, and tune $h$ to realize a 2+1D Ising transition. We find this choice of $(V_0, V_1)$ has a smaller finite size effect. By inverting the equations in (7), we have

$$
\begin{aligned}
g_0 &= \frac{4s+1}{2s+1} V_0 + \frac{4s-1}{2s+1} V_1, \\
g_1 &= \frac{4s-1}{s} V_1,
\end{aligned}
\tag{17}
$$

so this corresponds to the region $g_0, g_1 > 0$.

## 3.2 Finite-size-scaling

To look for a phase transition, we begin with the order parameter for the Ising phase transition, which in the Landau Level basis (see Appendix A) is given by

$$M = \sum_{m=-s}^{s} c_m^{\dagger} \sigma^z c_m. \tag{18}$$

In Figure 1(a), we have plotted the Binder cumulant, given by

$$U_4 = 1 - \frac{\langle M^4 \rangle}{3 \langle M^2 \rangle^2}, \tag{19}$$

and we see a crossing in the vicinity of $0.20 - 0.21$, that drifts to larger couplings with larger $N = 2s + 1$. With this evidence of there being a quantum phase transition, we can find further evidence that the phase transition is in the Ising universality class by assuming that $\Delta_\sigma$ is equal to 0.518, as is known for the universality class [23], and checking the the magnetization data, as seen in Figure 1(b). Here we see a good crossing for $N = 10, 12, 14$, which is consistent with the choice of $\Delta_\sigma$. The crossing is at a larger $h$ than the Binder cumulant crossing, that is because the Binder cumulant suffers from larger finite size effects. Similar finite size effects have also been observed in the two-layer model [1].

To see that the data is consistent with both $\eta (= 2\Delta_\sigma - 1)$ and $\nu (= 1/(3 - \Delta_\epsilon))$, critical exponents in the 3D Ising universality class, we have performed a data collapse to a universal scaling function, assuming that $\langle M^2 \rangle / (\sqrt{N})^{4 - 2\Delta_\sigma}$ has a functional form of $f_0 + f_1 x + f_2 x^2 + f_3 x^3$, where $x = (h - h_c)(\sqrt{N})^{3 - \Delta_\epsilon}$. In addition to the fit parameters, we pay attention to the quantity $\chi^2/\text{dof}$, with numerator $\chi^2$ given by

$$\chi^2 = \frac{\sum_n (O_n - F_n)^2}{\sigma_n^2}, \tag{20}$$

where $O_n$ is a measurement, $F_n$ is the expected value from the fit, and $\sigma_n$ is the variance for the measurement. The quantity $\chi^2/\text{dof}$ is reduced by the number of degrees of freedom (dof) for the fitted data, which is the number of measurements minus the number of free parameters. A good fit that captures the features of the data but does not overfit is close to 1. When we fix $\Delta_\sigma = 0.518$ and $\Delta_\epsilon = 1.41$ and leave the other five parameters free, we get a fit for the data $N = 10, 12, 14$, with $\chi^2/\text{dof} = 1.305$ and estimate for the critical coupling of $h_c = 0.2129(8)$, as seen in Figure 1(c). If instead we fix $h = 0.21$, as suggested by the Binder ratio, and leave six parameters including the critical exponents free, a fitting ($\chi^2/\text{dof} = 1.491$) gives us $\Delta_\sigma = 0.49(2)$ and $\Delta_\epsilon = 1.28(6)$, values consistent with Ising universality for these relatively small system sizes.

## 3.3 Dimensionless two-point correlator

To take the advantage of fuzzy sphere regularization, below we compute CFT dimensionless two-point correlators on a sphere [9] at equal time,

$$\begin{aligned}
G_{\phi\phi}(\theta) &= \langle \phi(\theta = \varphi = 0)\phi(\theta, \varphi = 0) \rangle \\
&= \frac{1}{(2\sin(\theta/2))^{2\Delta_\phi}},
\end{aligned} \tag{21}$$

where $\phi$ is a CFT primary operator, and $(\theta, \varphi)$ are the spherical coordinates specifying the positions of the two operators. We mainly focus on the lowest $\mathbb{Z}_2$-odd primary $\sigma$, which can

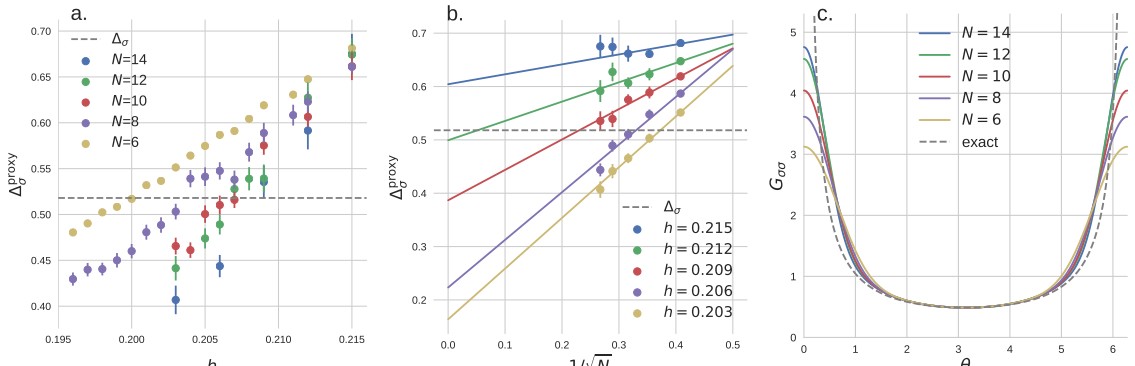

Figure 2: Data from CFT dimensionless two-point correlators. (a) Values of $\Delta_\sigma^{\text{proxy}}$ from $G_{\sigma\sigma}$ using the equal-time correlators. The crossing of the $\Delta_\sigma^{\text{proxy}}$ values through the $\Delta_\sigma = 0.518$ critical exponent value occurs at larger $h$-values as $N$ increases, consistent with the finite-size-scaling drift that was observed earlier. (b) Extrapolation to infinite lattice size $\Delta_\sigma$ from $G_{\sigma\sigma}$ correlation functions for different values of $h$ and $N = 6, 8, 10, 12, 14$. Linear fits suggest that, from this dataset, $h = 0.212$ is closest to criticality. (c) Plotting of the angle-dependence of the $G_{\sigma\sigma}$ correlator with data from $N = 6, 8, 10, 12, 14$ at $h = 0.212$.

be well approximated by the UV operator $n^z$ [8,9], up to a non-universal normalization (say $\sqrt{A}$) and higher order corrections $O(1/\sqrt{N})$ from operators with higher scaling dimensions. So we can first compute the equal-time two-point correlator,

$$
\begin{aligned}
f(\theta) &= \langle n^z(\theta = \varphi = 0) n^z(\theta, \varphi = 0)\rangle \\
&= \sum_{l=0}^{2s} \bar{Y}_{l,m=0}(\theta, 0) Y_{l,m=0}(0,0) \left\langle n_{l,0}^z n_{l,0}^z \right\rangle,
\end{aligned}
\tag{22}
$$

and then

$$
G_{\sigma\sigma}(\theta) = A f(\theta) + O\left(1/\sqrt{N}\right),
\tag{23}
$$

where $A$ is a nonuniversal number. Because we have an explicit expression for $f$, we know the exact values for $f(\theta = \pi)$ and $f''(\theta)|_{\theta=\pi}$, where the derivatives are taken in $\theta$. Then by assuming that $G_{\sigma\sigma}$ has the critical scaling form of (21), we can solve the following system,

$$
\begin{aligned}
A f(\pi) &= 1/(2\sin(\pi/2))^{2\Delta_\sigma^{\text{proxy}}}, \\
A f''(\theta)|_{\theta=\pi} &= \frac{\partial^2}{\partial\theta^2}\left(1/(2\sin(\theta/2))^{2\Delta_\sigma^{\text{proxy}}}\right)\Big|_{\theta=\pi}.
\end{aligned}
\tag{24}
$$

$\Delta_\sigma^{\text{proxy}}$ is a number that will extrapolate to the universal $\Delta_\sigma$ at the critical point as $N \to \infty$. Second derivatives are used for the second equation in (24) because the first derivatives in $\theta$ are zero for both $f$ and critical $G_{\phi\phi}$ at $\theta = \pi$.

Figure 2(a) shows the extracted $\Delta_\sigma^{\text{proxy}}$ values for different values of $h$ in the vicinity of the $h_c$ determined by finite-size-scaling. Here we see that the $\Delta_\sigma^{\text{proxy}}$ indeed crosses through the 3D Ising $\Delta_\sigma = 0.518$ value in this region, and furthermore we see that the $h$ at which this crossing occurs increases with system size $N$, which is consistent with the drift that we saw in the finite-size-scaling. Moreover, the drift appears to be slowing with increasing $N$, another consistency with finite-size scaling.

We can see more consistencies with finite-size-scaling from the results in Figure 2(b), which linearly extrapolate the values of $\Delta_\sigma^{\text{proxy}}$ as $N \to \infty$ for different values of $h$. Here we see that

for the infinite $N$ extrapolation, the $h = 0.212$ data is closest to the critical $\Delta_\sigma$, which is consistent with the earlier universal scaling fit of $h_c = 0.2129(8)$. We use this $h = 0.212$ data to show the calculation of $G_{\sigma\sigma}(\theta)$ as a function of $\theta$ for QMC data at $N = 6, 8, 10, 12, 14$ and how it approaches the exact expression as $N$ increases. The QMC data-derived expressions for $G_{\sigma\sigma}(\theta)$ are given in Figure 2(c) and are rescaled so that they are equal to the exact expression of $G_{\sigma\sigma}(\theta = \pi)$.

### 3.4 Energy gaps and state-operator correspondence

Next we turn to the state-operator correspondence [4, 5] on the sphere, namely, the scaling dimensions $\Delta_n$ are related to energy gaps by

$$\delta E_n = E_n - E_0 = \frac{v}{R}\Delta_n \,, \tag{25}$$

where $R$ is the radius of the sphere and $v$ is the model-dependent velocity of light.

While we are unable to get the full low lying energy spectrum directly using QMC, we are able to obtain energy gaps for the lowest lying states in each symmetry quantum number sector by using time-displaced correlation functions (see Appendix B). For an operator $O_S$ with the quantum number $S$, we have:

$$\langle O_S(\tau) O_S(0) \rangle = \sum_n a_n^2 e^{-\tau(E_{S,n} - E_0)} \,, \tag{26}$$

where $E_0$ is the ground state energy, $E_{S,n}$ represents the energies of eigenstates $|\psi_n\rangle$ in the quantum number sector $S$, $a_n$ is an operator $O_S$ and state $|\psi_n\rangle$ dependent non-universal factor. At long time $\tau \gg 1$, the lowest energy will dominate and can be extracted by fitting the exponential decay.

In the data that follows, we will use density operators $n_{l,m}^i$ to measure energy gaps in different quantum number sectors. Specifically,

1) $n_{l,m}^z$ can measure gaps in the $\mathbb{Z}_2$-odd, parity-even, and angular momentum (i.e. Lorentz spin) $l$ sector,

2) $n_{l,m}^x$ can measure gaps in the $\mathbb{Z}_2$-even, parity-even, and angular momentum $l$ sector,

3) $n_{l,m}^0$ can measure gaps in the $\mathbb{Z}_2$-even, parity-odd, and angular momentum $l$ sector.

Figure 3 shows QMC data at the critical point $h = 0.212$. The energy gaps are scaled such that the gap measured from $n_{l=2,m=0}^0$ is rescaled to 4, the scaling dimension of the lowest parity-odd descendent of the energy-momentum tensor, $\Delta_{\varepsilon_{\nu\rho\eta}\partial_\rho T_{\mu\nu}}$. In doing so, we find the gaps measured from other operators to be consistent with the scaling dimensions of primary and descendant operators of the 3D Ising CFT. The density operator ($n_{l=2,m=0}^x$) we measured does not seem to have an overlap with the state of stress tensor (with $\Delta_{T_{\mu\nu}} = 3$). Instead it gives the level-2 descendant of $\epsilon$ primary, i.e., $\partial_\mu \partial_\nu \epsilon$.

## 4 Conclusions

We have introduced a model that is amenable to using sign-problem free quantum Monte Carlo to simulate the $(2+1)-D$ transverse Ising model on a fuzzy sphere. Through finite-size scaling we have found data consistent with the model's phase transition being in the 3D Ising universality class, and we also have shown that we can recover the same critical exponents from the model's energy spectra, which is evidence of emergent conformal symmetry.

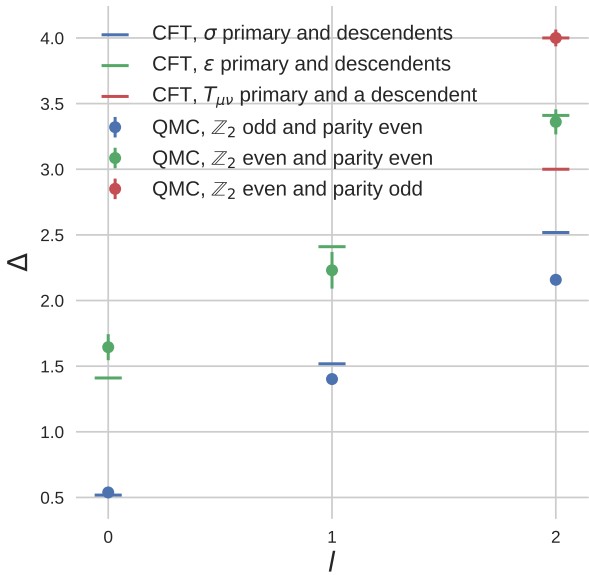

Figure 3: Comparison between 3D Ising CFT data and rescaled energy gaps for $N = 8$ and $h = 0.212$ measured by density operators. The energy gaps are rescaled by the factor that makes the lowest $\mathbb{Z}_2$-even, parity-odd gap at $l = 2$ equal to the lowest parity-odd descendent of the energy-momentum tensor, 4.0.

While these calculations are not competitive with ED and DMRG for small lattices, this work opens the door to larger scale calculations for models where there are too many degrees of freedom to make ED/DMRG calculations infeasible, or for when large sizes are desired for more accurate determination of critical exponents. Here, we were able to introduce an additional layer degree of freedom to avoid the sign-problem, and chose the interaction to disfavor spontaneous layer-symmetry breaking energetically. However, such instabilities cannot be ruled out in general. However, there are many interesting critical phenomena which naturally involve multiple flavors; one interesting target is the critical gauge theories proposed in Ref. [10].

# Acknowledgments

We thank Chao Han and Zheng Zhou for stimulating discussions.

**Funding information** Research at Perimeter Institute is supported in part by the Government of Canada through the Department of Innovation, Science and Industry Canada and by the Province of Ontario through the Ministry of Colleges and Universities. WZ is supported by National Natural Science Foundation of China (No. 92165102) and foundation of the Westlake University. JH was supported by the European Research Council (ERC) under grant HQMAT (Grant Agreement No. 817799), the Israel-US Binational Science Foundation (BSF), and by a Research grant from Irving and Cherna Moskowitz. FG acknowledges financial support through the German Research Foundation, project-id 258499086 - SFB 1170 'ToCoTronics'. This work used Bridges 2 at the Pittsburgh Supercomputing Center through allocation PHY170036 from the Advanced Cyberinfrastructure Coordination Ecosystem: Services & Support (ACCESS) program, which is supported by National Science Foundation grants #2138259, #2138286, #2138307, #2137603, and #2138296.

# A  Order parameter

The Ising order parameter is $n^z$, and for QMC simulations we measure the two-point correlation function,

$$
\begin{aligned}
\langle n^z(\theta,\varphi)n^z(\theta',\varphi')\rangle &= \sum_{l,m,l',m'} \langle n^z_{l,m}n^z_{l',m'}\rangle Y_l^m(\theta,\varphi)Y_{l'}^{m'}(\theta',\varphi') \\
&= \sum_{l,m}(-1)^m \langle n^z_{l,0}n^z_{l,0}\rangle Y_l^m(\theta,\varphi)Y_l^{-m}(\theta',\varphi').
\end{aligned}
\tag{A.1}
$$

The last equation comes from the conservation of angular momentum,

$$
\begin{aligned}
\langle n^z_{l,m}n^z_{l',m'}\rangle &= \begin{pmatrix} l & l' & 0 \\ m & m' & 0 \end{pmatrix} O_l \\
&= \delta_{l,l'}\delta_{m,-m'} \begin{pmatrix} l & l & 0 \\ m & -m & 0 \end{pmatrix} \langle n^z_{l,0}n^z_{l,0}\rangle / \begin{pmatrix} l & l & 0 \\ 0 & 0 & 0 \end{pmatrix} \\
&= (-1)^m \delta_{l,l'}\delta_{m,-m'} \langle n^z_{l,0}n^z_{l,0}\rangle.
\end{aligned}
\tag{A.2}
$$

Therefore, we need to evaluate $\langle n^z_{l,0}n^z_{l,0}\rangle$ for each $l$. To do the finite-size-scaling, we calculate the order parameter $\langle M^2\rangle$, with $M = \int d\Omega\, n^z(\theta,\varphi)$,

$$
\begin{aligned}
\langle M^2\rangle &= \int d\Omega d\Omega' \langle n^z(\theta,\varphi)n^z(\theta',\varphi')\rangle \\
&= \sum_{l,m}(-1)^m \langle n^z_{l,0}n^z_{l,0}\rangle \int d\Omega d\Omega'\, Y_l^m(\theta,\varphi)Y_l^{-m}(\theta',\varphi') \\
&= 4\pi \langle n^z_{0,0}n^z_{0,0}\rangle.
\end{aligned}
\tag{A.3}
$$

The last equation comes from $\int d\Omega Y_l^m(\theta,\varphi) = \sqrt{4\pi}\delta_{l,0}\delta_{m,0}$. Using Wigner-3j identities, we find that,

$$
\begin{aligned}
n^z_{0,0} &= (2s+1)\sqrt{\frac{1}{4\pi}}\sum_{m_1=-s}^{s}(-1)^{3s+m_1}\begin{pmatrix} s & 0 & s \\ -m_1 & 0 & m_1 \end{pmatrix}\begin{pmatrix} s & 0 & s \\ -s & 0 & s \end{pmatrix} c^\dagger_{m_1}\sigma^z c_{m_1} \\
&= \sqrt{\frac{1}{4\pi}}\sum_{m_1=-s}^{s} c^\dagger_{m_1}\sigma^z c_{m_1}.
\end{aligned}
\tag{A.4}
$$

Therefore, the order parameter $M^2$ is

$$
\langle M^2\rangle = \sum_{m_1,m_2=-s}^{s}\langle (c^\dagger_{m_1}\sigma^z c_{m_1})(c^\dagger_{m_2}\sigma^z c_{m_2})\rangle.
\tag{A.5}
$$

# B  Extracting energy gaps

In projector QMC, we are able to get the energy gap between the first excited state in symmetry sector $S$ and the ground state in the following way. Assuming a trial wavefunction, $|\psi_0\rangle$, an operator that creates overlap between the states in symmetry sector $S$ and the ground state, $O_S$, and a complete set of states $\sum_n |n\rangle\langle n|$, where $|n\rangle$ is an eigenstate with energy eigenvalue

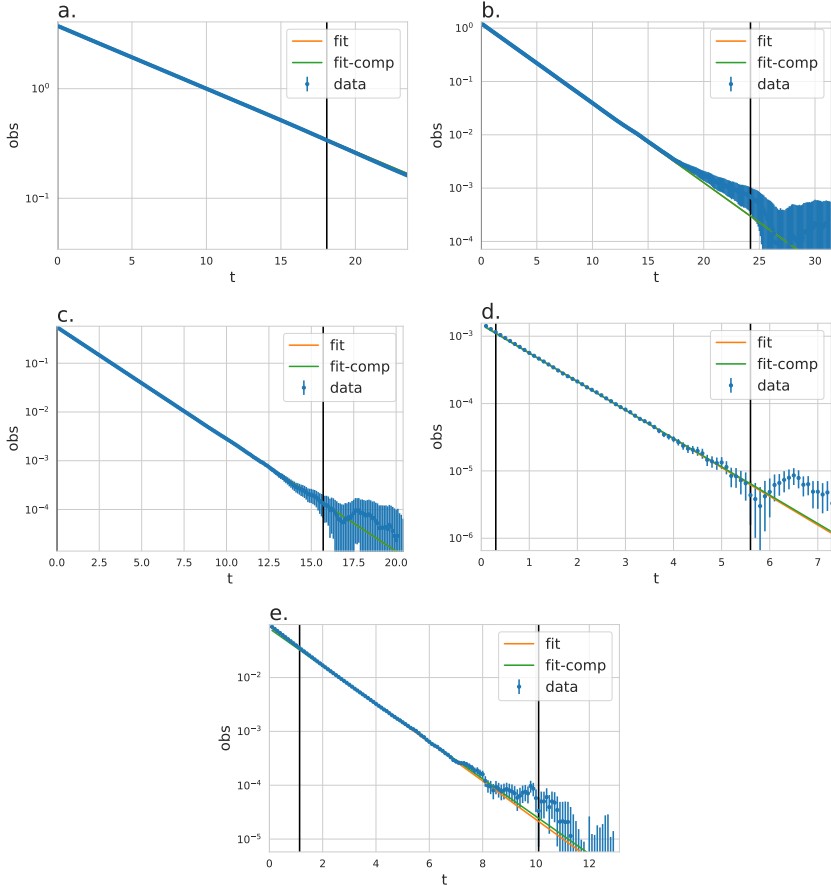

Figure 4: Fits to one exponential using semilog plots. The vertical lines show the locations of the endpoint and startpoints for the data. Plots (a), (b), and (c) show the data from $n^z_{0,0}$, $n^z_{1,0}$, and $n^z_{2,0}$, respectively. Plot (d) shows the data from $n^0_{2,0}$ and plot (e) shows the data from $n^x_{2,0}$.

$E_n$, we have that

$$
\begin{aligned}
\langle O_S(\tau) O_S(0)\rangle &= \frac{\langle \psi_0 | e^{-(\beta-\tau)H} O_S e^{-\tau H} O_S | \psi_0\rangle}{\langle \psi_0 | \psi_0\rangle} \\
&= \sum_n \frac{\langle \psi_0 | e^{-(\beta-\tau)H} O_S | n\rangle \langle n | e^{-\tau H} O_S | \psi_0\rangle}{\langle \psi_0 | \psi_0\rangle} \\
&= \sum_n \frac{\langle \psi_0 | O_S | n\rangle \langle n | O_S | \psi_0\rangle}{\langle \psi_0 | \psi_0\rangle} e^{-\beta E_0} e^{-\tau(E_n - E_0)}.
\end{aligned}
\tag{B.1}
$$

This term has contributions from all eigenstates that have the symmetry $S$. The higher energy states will have gaps that will be suppressed relative to the smallest energy gap, and so we can approximate

$$
\langle O_S(\tau) O_S(0)\rangle = C_1 e^{-\tau(E^0_S - E_0)} + C_2 e^{-\tau(E^1_S - E_0)},
\tag{B.2}
$$

where $E^0_S$ and $E^1_S$ are the lowest energy and second lowest energy corresponding to states in symmetry sector $S$, respectively.

In practice, we found that in order to extract $E^0_S$, sometimes a fit to the two exponentials with prefactors $C_1$ and $C_2$ in (5) is necessary, but sometimes a fit to a single exponential (which assumes $C_2 = 0$) is more appropriate. The procedure for fitting to one versus two exponentials involves the following steps:

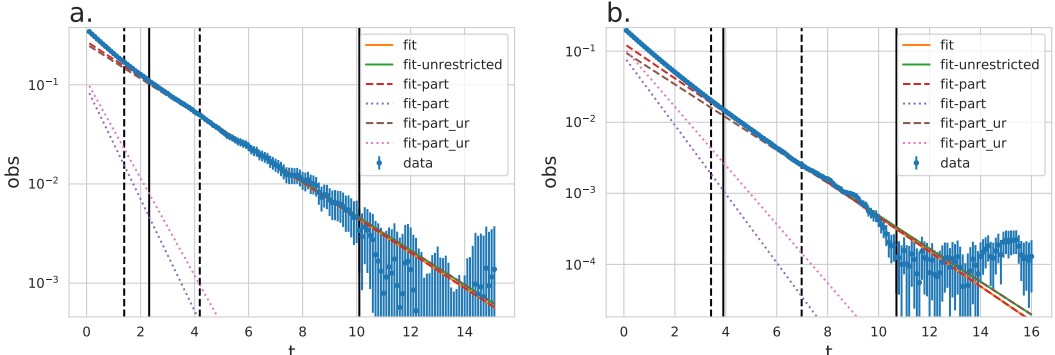

Figure 5: Fits to two exponentials using semilog plots. The vertical solid lines show the locations of the endpoint and midpoint for a restricted fit–where the $C_1$ exponential and the $C_2$ exponential are fitted separately but iteratively using information from previous results to arrive at a final answer. The dashed lines give an interval of midpoints that were tested in order to estimate the error bar. The solid diagonal lines give the two-exponential restricted and unrestricted fits ("fit" and "fit-unrestricted"). The dashed diagonal lines are fits for each of the two exponentials, both for restricted ("fit-part") and unrestricted ("fit-part_ur") fits. Plot (a) shows the data from $n_{0,0}^x$, and plot (b) shows the data from $n_{1,0}^x$.

1. Find the $\tau$ interval where the data is distinguishable from zero according to error bars. The largest time in this interval is the initial "endpoint" guess.

2. Initially guess that the "midpoint" in time–where one exponential versus the other exponential dominates–is 30% of the full time interval.

3. Test a fit to a single exponential–if the initial data point is smaller than the $t = 0$ value for the fitted function, gradually adjust the midpoint and endpoint guesses down until this is not the case.

4. If the initial data point is still within errors of the $t = 0$ value for the single-exponential fitted function, fit the data to a single exponential. If not move on to a two-exponential fit and then revise the midpoint guess such that the value of the larger exponential in the fit is negligible compared to that of the smaller exponential at the midpoint.

For the operators $n_{l,0}^z$, $n_{2,0}^0$, and $n_{2,0}^x$, the fit to a single exponential ends up being more appropriate. Figure 4 shows the fits to a single exponential (in the cases of $n_{2,0}^2$ and $n_{2,0}^x$, we chose a single exponential because there was very little small $\tau$ data to fit to a higher exponential). However, the $n_{0,0}^x$, $n_{1,0}^x$ observables require two exponentials. Figure 5 shows the two exponential fits for these operators at coupling $h = 0.212$ and $s = 3.5$.

For the two-exponential fits, we first use the midpoint data to iteratively fit one exponential at a time: midpoint to endpoint is the fit for the lower energy and then the startpoint to the midpoint is the fit for the higher energy. We alternate fitting one exponential versus the other while fixing the parameters of the nonfitted exponential according to the previous fit. This gives us the "restricted" fit listed as "fit" in Figure 5. We then use these fitted energy values as initial guesses for an "unrestricted" fit that fits both exponentials at once. This gives the "fit-unrestricted" in Figure 5. The value of the energy estimate is the mean of these restricted and unrestricted fit energies. Finally, we obtain error bars by performing fits to a single exponential

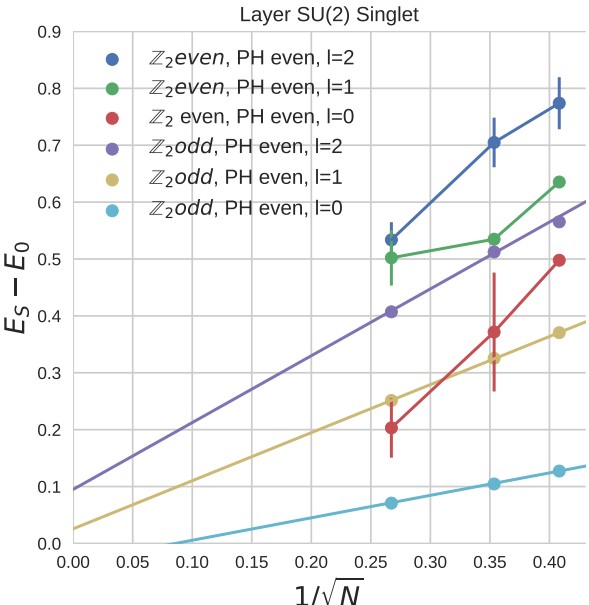

Figure 6: Data showing the energy gaps obtained from the $SU(2)$ singlet operators as a function of $1/\sqrt{N}$ for $N = 8, 10, 14$. This data is in the vicinity of the critical point at $V_1 = 0.1, V_0 = 0.5564, h = 0.2$. The gaps decrease with system size, as expected. Extrapolations are shown for the $\mathbb{Z}_2$-odd sector, but not the even sector since the error bars are so large.

for the smaller energy from a midpoint to the endpoint, where we calculate the midpoint as

$$\tau = -\frac{1}{(E_S^1 - E_S^0)\ln\left(\frac{\epsilon C_1}{C_2}\right)}, \tag{B.3}$$

where $\epsilon$ is a small number representing the time when the value of $C_2 e^{-\tau(E_S^1 - E_0)}$ is $\epsilon$ times $C_1 e^{-\tau(E_S^0 - E_0)}$. We take a range of $\epsilon \in \{0.01, 0.1\}$ to fit $E_S^0$ and use this range of $E_S^0$ values to estimate the error for the energy. The boundaries of this range of midpoints are given by the dashed vertical lines in Figure 5.

## C   Finite size scaling of energy gaps

Because the QMC model studied has an additional $SU(2)$-layer symmetry, one check to make is whether the degrees of freedom in the layer $SU(2)$ non-singlet representations are gapped at the phase transition. All the operators in the Hamiltonian are of the form $\sigma^i \tau^0$, which are the singlets of the layer $SU(2)$ symmetry. Figure 6 shows the energy gaps obtained from these operators as a function of $1/\sqrt{N}$ at $h = 0.2$, which is in the vicinity of the critical point. All gaps are decreasing with system size and the gaps for the $\mathbb{Z}_2$-odd sector seem to be trending linearly towards the origin, as required by the state-operator correspondence. The $\mathbb{Z}_2$-even sector gaps are also decreasing with increasing system size, but they have larger error bars due to interference with other higher energy descendants in their spectrum, and the details of their fits are given in Appendix B of the Supplementary Material. On the other hand, the layer $SU(2)$ non-singlet, e.g., layer triplet gaps should be finite in the thermodynamic limit. Figure 7 shows layer triplet energy gaps measured by $\sigma^i \tau^{x,y,z}$, as a function of $1/\sqrt{N}$, and from here we see that these excitations appear to be gapped.

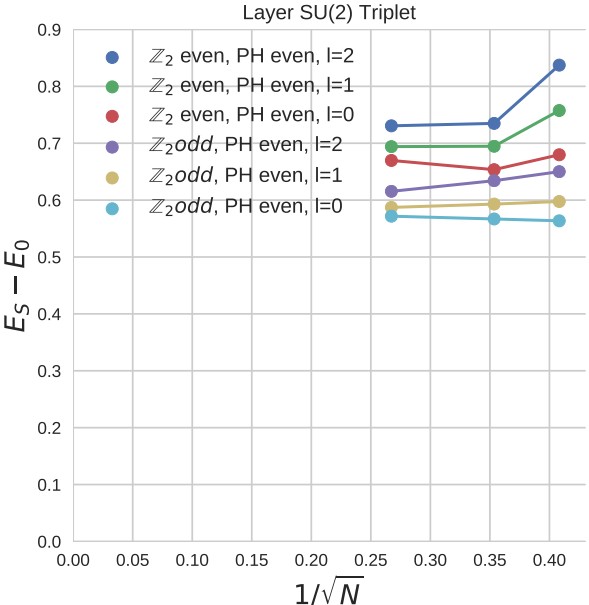

Figure 7: Data showing the energy gaps obtained from operators of the form $\sigma^i \tau^z$ as a function of $1/\sqrt{N}$ for $N = 8, 10, 14$ (operators of the form $\sigma^i \tau^{x,y}$ would give the same states, making this the $SU(2)$ triplet symmetry class). This data is in the vicinity of the critical point at $V_1 = 0.1, V_0 = 0.5564, h = 0.2$. These energies appear to be gapped out.

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
