# Peer review of "Quantum Monte Carlo Simulation of the 3D Ising Transition on the Fuzzy Sphere"

_SciPost Physics Core, doi:SciPost Phys. Core 7, 028 (2024)_

## Round 1 · Referee Report · Anonymous (Referee 1) · 2024-3-5

Strengths

1- the topic is very interesting 2- the mathematical steps are clear and well documented

Weaknesses

1- quite specialised 2- not very clear how to generalise the method

Report

The paper deals with a very interesting problem of simulating phase transitions on the fuzzy sphere using quantum Monte Carlo. Recently fuzzy sphere regularization has received attention in several branches of theoretical physics so it is timely to address and resolve potential issues. The authors propose a novel approach to the simulation by introducing flavor index to the fermions. Remarkably, this approach does not display any sign problem. The authors show it in the case of the 3D Ising model. While the paper is well written and provides evidences of the effectiveness of quantum Monte Carlo in this setting, it would be nice if the authors will clarify in more details how general the method of adding flavour to fermions related to the sign problem. In particular how it can be used in more general phase transitions.

Requested changes

1-explain the limitations of the approach
2-discuss the generality of the method

  • validity: good
  • significance: high
  • originality: high
  • clarity: good
  • formatting: good
  • grammar: excellent

Author:  Emilie Huffman  on 2024-04-09  [id 4401]

(in reply to Report 1 on 2024-03-05)

We thank the referee for taking the time to read our manuscript and provide a useful report. As we detail below, we have addressed the referee's comments appropriately and feel confident that the modified manuscript can now be published by SciPost.

  1. We have added a discussion of the numerical limitation, i.e., the scaling of compute time with system size, below Eq. [16]. Due to the rank of the operators that couple to the auxiliary fields, the computational effort scales as N^4 instead of the usual N^3, which can be achieved for conventional lattice models. However, we stress that this method still scales polynomially with size, in contrast to exact diagonalization and DMRG. Hence, the QMC remains the method of choice when many fermionic degrees of freedom are studied, e.g., for studying critical gauge theories with multiple fermion flavors.

  2. We have added a more detailed discussion of the guiding design principle for the additional flavor in the last paragraph of Sec. 2.3. It might indeed be interesting to study how generically one can use this trick to study universal aspects of critical phenomena. However, the purpose of this paper is not to show that introducing an additional flavor can always provide a sign-free model to study any universality class of interest. Instead, we introduce the possibility of using the QMC method on the fuzzy sphere, which has never been done before in a way that exploits the unique features of it. For example, the state-operator correspondence on the fuzzy sphere gives rise to unambiguous fingerprints of conformal symmetry in the spectrum (Fig. 3). Note that this cannot be achieved on a tight-binding Hamiltonian on a 2D lattice with periodic boundary conditions, nor by conformal bootstrap as the conformal symmetry that assumes conformal symmetry from the beginning. Similarly, the conformal symmetry highly constrains the functional form of (equal-time) correlation functions which allow a fit-free extraction of critical exponents/operator dimensions (Fig. 2c) We agree with the referee that this paper mostly focuses on the Ising universality class to benchmark the method. In response to their comments, we have also added a discussion regarding the "generality of the method" to the conclusion. For the Ising model, we needed an additional flavor degree of freedom to avoid the sign problem. However, many interesting universality classes can be studied using sign-free model Hamiltonians without the need to introduce artificial flavor degrees of freedom! Examples are the critical gauge theories that have been proposed to describe deconfined quantum critical points.

Attachment:

RevisedFuzzySphereQMC_hQfbJu0.pdf

---

## Round 1 · Referee Report · Anonymous (Referee 2) · 2024-3-22

Strengths

1 - The presented method shows promise to circumvent the fermion sign problem for a new class of models.
2 - The paper presents a QMC implementation of the recently introduced fuzzy sphere regularization and takes care of benchmarking of known results.
4 - The paper is clearly written and self-contained concerning the technical steps.

Weaknesses

1 - It is not clear whether the paper meets the criterion of presenting new research results that significantly advance the current understanding of the field as it is limited to one benchmarking result. 2- It does not become clear whether the method can substantially advance open problems in the field. 3 - The benchmarking seems to only allow for a check of consistency with the Ising universality class, but not to make actual precision predictions.

Report

This manuscript presents an interesting sign-free implementation of the quantum Monte Carlo method employing the recently proposed fuzzy sphere regularization. It studies the quantum phase transition and critical behavior as a function of the transverse field and shows that the behavior is compatible with 3D Ising universality. The authors claim that this strategy can be used to generalize to other models and simultaneously avoid the notorious sign problem.

The content of the paper is interesting, it is well written, self-contained, and the results are presented clearly. The benchmark for Ising criticality presents an interesting proof of concept for the method. However, I'm missing a concrete example and actual results that go beyond the benchmark example.

If this is clearly beyond the scope of the current work, it should be argued more clearly why -- despite the limitation to the benchmarking result -- the presented result meets the acceptance criteria, i.e., that the work significantly advances the current understanding of the field and that it not only reproduces known results. A better alternative would be to find and calculate an actual example that goes beyond established results.

Requested changes

1 - Above Eq. (3), the authors mention that they consider the limit that the kinetic energy is much larger than the interaction energy. It should be discussed how restrictive this constraint and how it affects the presented results.
2 - The authors should discuss more clearly, whether consistency is all that can be expected to be achieved within the new method or whether there is a realistic perspective to achieve quantitative precision results, e.g., on critical exponents.
3 - If the answer to the previous question is that quantitative results can be achieved, the authors should discuss how expensive it would be to do this for their current model or -- even better -- provide such quantitative precision results.
4 - I'm worried that the fitting in Sec. 3.2 may be affected by some sort of overfitting due to the presence of many parameters. The authors should comment on that.
5 - The quantity chi^2 should be introduced/defined. It might not be obvious to everyone outside the numerical community.
6 - The authors should consider to discuss the possible sources/implications of corrections to scaling and how they might affects their results.
7 - An interesting non-trivial benchmark could be the critical behavior of Dirac fermions, where high precision estimates are available from the conformal bootstrap community, but the QMC results are not fully settled yet. Maybe the authors could consider this.

  • validity: good
  • significance: good
  • originality: high
  • clarity: high
  • formatting: good
  • grammar: excellent

Author:  Emilie Huffman  on 2024-04-09  [id 4400]

(in reply to Report 2 on 2024-03-22)

We would like to thank the referee for taking the time to read our manuscript and for providing a report that helped us to improve this paper. We are happy to hear that the referee finds our work interesting and for rating the originality of this study as 'High'.

The target of the present work is to introduce a new method, namely the fuzzy sphere QMC, and the feasibility of this approach is demonstrated via the benchmark on the Ising model. We believe this approach will advance the coming studies on more challenging problems, which we leave for the future work. We do think, as a paper introducing new techniques, it should not influence the quality of the current paper.

First, we agree that high-precision results for critical exponents have already been obtained for the Ising universality class by various methods such as the conformal bootstrap, and QMC for spin lattice models. Instead of competing with these already very efficient methods for the Ising universality class, the fuzzy sphere setup allows us to examine the proposed emergent conformal symmetry at the critical point, and further make use of the conformal symmetry to compute universal properties such as critical exponents.

Second, in a previous study introducing the fuzzy sphere setup, we have used both exact diagonalization and DMRG to identify the Ising critical point and analyze the Hamiltonian spectrum at criticality to verify the emergent conformal symmetry. However, both methods struggle with increasing fermion degrees of freedom, due to the exponential compute cost. Here, we introduce a polynomial algorithm to overcome these limitations and benchmark the new method using the Ising criticality.

Finally, the polynomial scaling allows us to study models with larger local Hilbert spaces and the plethora of interesting quantum critical phenomena therein. We agree with the referee that it would be rewarding to study critical Dirac fermions, for example. However, this is left for future work.

Hence, we strongly believe that the presented research is already an important result and meets the acceptance criteria of SciPost. A detailed response to the list of requested changes can be found below. With these changes we believe the manuscript is ready for publication.

Requested changes:

  1. We agree that it is useful to clarify the projection. As given in the manuscript, we have H = H_{kin} + H_{int}, with H_{kin} giving rise to Landau levels and H_{int} the Hamiltonian of physical interest. The level spacing of the Landau levels is controlled by the cyclotron frequency $\omega_c$, and we consider a half-filled lowest Landau level (LLL). In general, the interaction scale could induce excitations to higher Landau levels, however, this can be avoided by taking the limit $\omega_c \rightarrow \infty$. Hence, the projection of H_int, encoding the physics of interest, to the LLL is not posing any constraint. We have added a couple of sentences above equation (3) to clarify this.

2.,3. To reiterate the argument from above, the purpose of this work is not to produce high-precision critical exponents for the Ising universality class. Instead, we are introducing a sign-free QMC algorithm with a polynomial computational effort. While in principle, one can produce relatively precise exponents, we would like to highlight that this setup also enables us to analyze the existence of an emergent conformal symmetry at criticality, especially for models with larger local Hilbert spaces. However, applying the new algorithm to such models to study, e.g., critical gauge theories, relevant for deconfined quantum critical points, is left for future studies. Here, we have shown that the algorithm actually works by benchmarking it using the Ising model. To address the referee's comments, we have added a discussion of the numerical complexity below Eq. (16), and discuss future directions in the conclusion.

4.,5.,6. We have added the definition of the chi^2 to the paper, see new Eq. (20). While there are seven parameters, we also have thirty-two data points in our fit, so overfitting should not be a big concern. Note that we are not trying to extract high-precision critical exponents from the scaling collapse, which are known to be prone to finite size scaling corrections, but we are rather giving evidence that the critical point of the fuzzy sphere model belongs to the Ising universality class. We believe the current system sizes are sufficiently large for this statement as shown by the quality of the scaling collapse in Fig. 1(c).

7., We thank the referee for this interesting comment and agree that this would be an interesting future direction. However, it is still an open nontrivial question how to properly describe Dirac fermions on a fuzzy sphere. A more directly accessible follow-up is to study critical gauge theories, as already mentioned in the conclusion. However, we believe that this would be a sufficiently different and new result to be left for a future work.

We have attached a revised version of the manuscript for the reviewer to examine.

Attachment:

RevisedFuzzySphereQMC_BUzhntM.pdf

---

## Round 1 · Referee Report · Anonymous (Referee 3) · 2024-3-25

Strengths

1 -The new four-flavor fermions model overcomes the sign problem for QMC of the fuzzy-ball regularization approach to 3D Ising criticality.
2 - The paper is well-written and provides the necessary technical details of the analytical derivations.
3 - The numerical part provides a careful finite-size scaling analysis over several measures.

Weaknesses

1 - The performance of the current model compared to the traditional MC approach, such as the lattice Ising model, remains unclear.
2 - The proposed model seems to lack an intuitive physical interpretation linking it to actual physical systems. While it serves as a numerical tool for determining critical exponents, its complexity may hinder future analytical progress.

Report

GPT
The paper presents a new four-flavor fermions model as an extension of the authors' recent work on the fuzzy-ball regularization approach to 3D Ising criticality. In contrast to the existing model, which consists of two-flavor fermions and suffers from the sign problem in Quantum Monte Carlo (QMC) simulations, the new model introduces an extra layer of SU(2) symmetry that protects the model from the sign problem. The paper is well-written, offering the necessary technical details for the analytical derivations.

The numerical section provides a thorough finite-size scaling analysis across several measures. However, the current model's performance relative to traditional Monte Carlo approaches, such as the lattice Ising model, remains unclear. Including a computational complexity analysis, even if only numerical, would significantly enhance the model's validity.

Additionally, the proposed model appears overly complex when attempting to link it to actual physical systems. Although it serves as a numerical tool for determining critical exponents, its complexity could impede future analytical advancements.

Requested changes

1 - Include a computational complexity analysis.
2 - The parameters V1 = 0.1 and V2 = 0.5564 appear to be fine-tuned, particularly for V2, which is chosen with four significant figures. Although the authors mention that this choice minimizes finite-size effects, it would be valuable to examine the results for other parameters.

  • validity: good
  • significance: high
  • originality: high
  • clarity: high
  • formatting: good
  • grammar: excellent

Author:  Emilie Huffman  on 2024-04-09  [id 4399]

(in reply to Report 3 on 2024-03-25)

Requested changes:

  1. We thank the reviewer for this suggestion and have added a discussion of the N^4 scaling under equation 16 in the QMC simulations section of Results.

  2. At the Ising critical point, the finite-size effect can be significantly reduced by tuning away the irrelevant operator epsilon’ (with scaling dimension ~3.83). In the present work, we tune the interaction parameters to reach this goal, i.e. g1=0.4 and g0=1.3 from the exact diagonalization. By changing these numbers to the conventions of QMC parameters, it gives the repeating decimals V0=0.61818181... and V1=0.1111111... We then scaled them so that V1=0.1, giving the repeating decimal V0=0.55636363... In principle, the Ising transition and its typical critical exponents should not depend on the specific parameters in the model setup in the UV limit. But the RG flow distances to the IR fixed point are distinct, if choosing the different parameters in the UV limit. So when we tune away from the "sweet point" as shown above, the finite-size effect will become significant and one needs much heavier computation on larger system sizes to reach the critical phenomena.

We have attached an updated version of the manuscript.

Attachment:

RevisedFuzzySphereQMC.pdf

---

## Round 2 · Referee Report · Anonymous (Referee 2) · 2024-4-11

Report

In response to my earlier report, the authors have responded to my comments and revised their manuscript accordingly. In particular, they have dispelled my doubts about the suitability concerning SciPost's criteria of acceptance by their clarifications. It's a good paper with a clear scope, interesting new results, and a nice perspective for future applications.

I also find their answers to my other comments persuasive and I therefore recommend publication of the manuscript in SciPost, now.

Recommendation

Publish (meets expectations and criteria for this Journal)

---

## Round 2 · Referee Report · Anonymous (Referee 1) · 2024-4-16

Report

I would like to thank the authors for carefully answering my comments and for implementing them into the text. The manuscript can be published as it is.

Recommendation

Publish (easily meets expectations and criteria for this Journal; among top 50%)

---

## Round 2 · Referee Report · Anonymous (Referee 3) · 2024-4-23

Report

The revised manuscript has successfully addressed my concerns related to computational complexity and parameter fine-tuning. I recommend it for publication as it currently stands.

Recommendation

Publish (easily meets expectations and criteria for this Journal; among top 50%)

---

## Round 2 · Author Response

Dear Editor,

We thank you for considering our manuscript for publication in SciPost and for soliciting reports
from these referees. We have responded to each referee report and resubmitted a revised manuscript. With these changes and our responses below, we hope our manuscript is suitable for publication in SciPost.

Best Regards,
Authors

---

## Round 2 · List of Changes

• added a discussion of the N^4 scaling under equation 16 in the QMC simulations section of Results
  • added a couple of sentences above equation (3) to clarify the Landau projection
  • added the definition of the chi^2 to the paper, see new Eq. (20)
  • added a more detailed discussion of the guiding design principle for the additional flavor in the last paragraph of Sec. 2.3
  • added more discussion of future directions in the conclusion
  • added an "int" subscript to equations (8) and (13)

---

## Editorial Decision

published